# ATFormer: A Learned Performance Model with Transfer Learning Across Devices for Deep Learning Tensor Programs

**Yang Bai,    Wenqian Zhao,    Shuo Yin,    Zixiao Wang,    Bei Yu**

Department of Computer Science and Engineering
The Chinese University of Hong Kong, Hong Kong SAR
{ybai,wqzhao,syin22,zxwang22,byu}@cse.cuhk.edu.hk

## Abstract

The training and inference efficiency of ever-larger deep neural networks highly rely on the performance of tensor operators on specific hardware platforms. Therefore, a compilation-based optimization flow with automatic tensor generation and parameter tuning is necessary for efficient model deployment. While compilation-based methods with performance models can provide dynamic and suitable code optimization, they suffer from a large design space exploration with rough measurement accuracy and poor transferability among different hardware platforms. This paper presents ATFormer, a simple yet efficient design with attention-inspired modules to accurately predict the performance of optimized operators by capturing global and long-range dependencies within a complete scheduling space. Compared with state-of-the-arts, ATFormer can predict the optimal implementation of tensor operators to reduce inference time with minimal effort on modern DNN benchmarks. Furthermore, ATFormer with pre-trained parameters can quickly adapt to different workloads and hardware via transfer learning.

## 1 Introduction

Recently, there has been a significant improvement in model performance for deep neural networks (DNNs) (He et al., 2016; Sandler et al., 2018; Shan et al., 2021; Devlin et al., 2019; Wu et al., 2019; Biten et al., 2019; Bello et al., 2019). However, this progress has been accompanied by a significant increase in the number of operators and, consequently, the computational complexity of DNNs. As a result, it has become increasingly challenging to efficiently deploy DNNs with optimized tensor programs on certain hardware accelerators like CPUs, GPUs and TPUs (Jouppi et al., 2017).

To overcome the limitations, mainstream search-based tensor compilers (Chen et al., 2018a; Zheng et al., 2020; Bai et al., 2021; Li et al., 2020; Fegade et al., 2021) are developed. These compilers

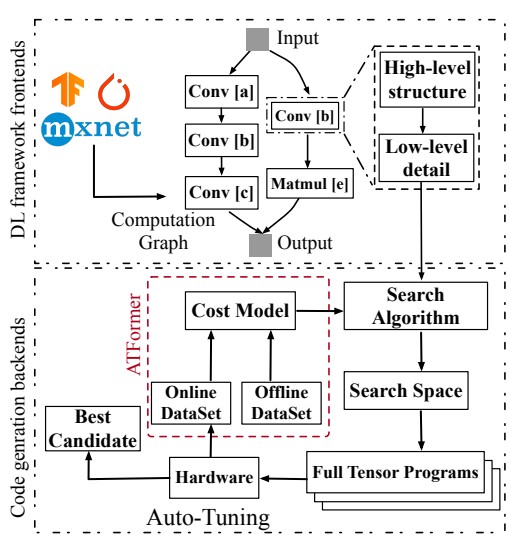

Figure 1: The overview of a search-based framework with computation graph, cost model, and search space.

automatically search for the optimal deployment configuration of each operator on increasingly heterogeneous platforms. Conducting on-device measurements is extremely time-consuming, making it impossible to place all the generated tensor programs on the target platform for measurement during the compilation process. Therefore, the prediction via an optimal cost model is crucial in reducing the time-consuming measurements during the compilation which can significantly improve search efficiency and quality.

Nevertheless, the existing cost models are capable of selecting nearly optimal configurations but suffer from excessively long optimization time. These long optimization times not only impede the deployment period but also raise concerns about the practicality of search-based compilers. Furthermore, statistic cost models trained on one hardware platform exhibit significant performance degradation on different hardware, making them unusable across different platforms. It is noteworthy that the execution times of tensor programs can vary signif-

icantly on different platforms due to domain gaps, making it challenging to deploy optimized models on multiple platforms. This is further compounded by the significant differences in the features extracted from various platforms. Even when extracted on GPUs, the feature's stability and performance cannot be guaranteed across different GPU architectures such as Volta, Turing, and Ampere. Therefore, additional engineering efforts are necessary to account for the differences in hardware architectures, resulting in a laborious and cumbersome feature extraction process.

To address these challenges, we propose a powerful yet simple approach that uses attention-inspired blocks to enhance the performance of cost models. These blocks can capture global and long-range dependencies among tensor program statements. Additionally, transferable features with pre-trained parameters are used to expedite search convergence across different hardware platforms. These techniques can be easily incorporated into existing search algorithms and improve efficiency in an end-to-end fashion. Our design, ATFormer, consistently outperforms popular DNN benchmarks, including small and large-scale models. Furthermore, our techniques enable cross-platform transfer learning, resulting in more efficient deployment.

The main contributions of this paper are the following: (i) We highlight the limitations of current auto-tuning frameworks. Existing tree-based performance models are insufficient for evaluating inference in a large search space and transferable knowledge is difficult to acquire across different platforms. (ii) A simple yet efficient design that utilizes attention-based blocks to explore the correlation between all innermost non-loop statements in a full tensor program, resulting in accurate prediction. (iii) Our approach enables rapid adaptation of performance tuning across various GPU platforms using pre-trained parameters on static datasets, not only in cross-operator but also cross-platform scenarios. Comprehensive experiments on modern DNN benchmarks and the large-scale TenSet (Zheng et al., 2021) demonstrate the consistent and superior performance of our method.

## 2 Background and Related Work

**Deep Learning Compiler.** Recently, the development of compiler-based optimization frameworks, such as Halide (Adams et al., 2019), TVM (Chen et al., 2018b), XLA (Sabne, 2020),

and TACO (Kjolstad et al., 2017), has progressed rapidly. These optimization schemes typically consist of two parts: DL framework frontends and code generation backends, as illustrated in Figure 1. The frontend converts an input model into a high-level graph-based intermediate representation (IR) and applies target-independent optimizations, such as operator fusion and data layout transformation. In the backend, target-dependent optimization passes, along with hardware features, further optimize the final performance. TVM (Chen et al., 2018a) is a state-of-the-art search-based tensor compiler that is widely used in academia and industry. Its auto-tuning aims to achieve performance comparable to hand-tailored libraries and has achieved promising results. TVM has two versions of auto-tuning: AutoTVM (Chen et al., 2018c) and Ansor (Zheng et al., 2020). While AutoTVM is a semi-automated framework that requires pre-defined manual templates, Ansor is more advanced and fully automated. However, both frameworks need to collect data on-the-fly during the search, resulting in an extremely long compilation time.

**Tree-based Performance Model.** Decision trees are frequently used in classification and regression problems. To enhance their performance, an ensemble learning approach is typically employed to reduce variance. XGBoost (Chen and Guestrin, 2016a) and LightGBM are powerful feature models in sequence modeling tasks. To achieve accurate prediction, a number of works, including (Chen et al., 2018c; Zheng et al., 2020; Ahn et al., 2020; Gao et al., 2021; Bai et al., 2021, 2023; Huang et al., 2023; Zhao et al., 2023), use XGBoost as the performance model during the tuning. AutoTVM extracts domain-specific features from a provided low-level abstract syntax tree (AST). During optimization, these features, which include loop structure information and generic annotations, are explored. Moreover, TreeGRU (Tai et al., 2015) recursively encodes a low-level AST into an embedding vector, which is mapped to a final predicted score within a fully-connected layer to enhance performance. Halide (Adams et al., 2019) builds regression models with hardware-specific features for auto-scheduling. TabNet (Arık and Pfister, 2020) uses sequential attention to select the most salient features to reason at each decision via a deep tabular architecture.

**DNN-based Performance Model.** In contrast, some recent approaches aim to reduce the impact of

search algorithms on final performance by utilizing more robust and powerful cost models. (Kaufman et al., 2020) and (Sun et al., 2022) employ graph neural networks to predict the latency of DNNs on TPUs. (Steiner et al., 2021) formulates the tuning process as a deterministic Markov Decision Process (Xiang et al., 2015) and solves it by learning an approximation of the value function. Tiramisu (Baghdadi et al., 2019) manually extracts 2534 features from the structure of AST, and forwards the AST as a computation stream to propagate features during the training. These models are trained effectively on a dataset with only a few thousand schedules using the hardware-dependent features crafted by heavy feature engineering techniques. However, complex feature engineering can become problematic in such cases. As hardware-specific features are difficult to transfer to a new platform, a learned performance model trained on one hardware platform typically performs poorly on another. This leads to an issue we call cross-hardware unavailability. Additionally, this approach cannot keep pace with the rapid development of new hardware, which further exacerbates the problem.

## 3 Methodology

### 3.1 Problem Formulation

We describe a DNN model as a computation graph and then define some important terminologies.

**Definition 1** (Subgraph). *Computation Graph $G$ is partitioned into a set of subgraphs $S$ based on the graph-level optimizer (Roesch et al., 2018).*

Each search task is extracted from an independent subgraph $S_i$ on a specific hardware platform $\mathbb{H}$. Thus, we define search task $Q$ as follows:

$$Q_{\mathbb{H}(S|G)} = \left\{ Q^1_{(S_1|G)}, Q^2_{(S_2|G)}, \ldots, Q^n_{(S_n|G)} \right\}, \tag{1}$$

where $n$ is the number of subgraphs in $G$. Note that each subgraph $S_i$ contains a computation-intensive operator $\sigma$ and $\sigma \in S_i$. Therefore, we use $Q^i_{(S_i|G)}$ to represent the $i-$th search task in $G$. Each subgraph $S_i$ has its own search space, which is determined by the input and output shapes, data precisions, memory layout, and the hardware platform. The search space is usually large enough to cover almost all kinds of tensor candidates.

**Definition 2** (Hierarchical Search Space). *A tensor program, denoted by $p$, represents an implementation of the subgraph using low-level primitives that*

---

**Algorithm 1** Search-based Framework

**Input:** Search space $\phi_1, \phi_2$ with operator $\sigma$ and setting $k$.
**Output:** Tensor program $p^*$ with best configuration $c^*$.
1: **while** nTrials < eachSubgraphTrials **do**
2:     $GS_1 \leftarrow$ GenerateHighSketch$(\phi_1, \sigma, k)$;
3:     $GS_2 \leftarrow$ Sampling$(GS_1, \phi_2, \sigma, k)$;
4:     P $\leftarrow$ EvolutionSearch$(GS_1, GS_2)$;
5:     **for** $p \in$ P **do**
6:         $c \leftarrow f(\eth(\phi_1, \phi_2|\sigma, k))$;
7:     **end for**
8:     nTrials $\leftarrow$ nTrials $+$ batchSize;
9: **end while**
10: $c^* \leftarrow$ best tensor program configurations;

---

*are dependent on the hardware platform. Each tensor program can be considered as a candidate in the search space. We define the hierarchical search space $\phi_{1,2}$, which decouples high-level structures $\phi_1$ from low-level details $\phi_2$, allowing for the efficient exploration of potential tensor candidates during the tuning process.*

Here, we can transform a tuning problem into an optimization problem that explores the potential tensor programs in a hierarchical search space.

**Problem 1.** *Given code generation function $\eth$, high-level structure generation parameters $\phi_1$, low-level detail sampling parameters $\phi_2$, computation-intensive operator $\sigma$ and operator setting $k$ (e.g., kernel size), our goal is to use $\phi_{1,2}$ to build a hierarchical search space and generate tensor program $p$ to achieve the optimal prediction score $y^*$ on a specific hardware platform $\mathbb{H}$.*

$$\begin{aligned} \phi^*_{1,2} &= \arg\max_\phi y, \\ y &= f_{\mathbb{H}}(\eth(\phi_1, \phi_2|\sigma, k)). \end{aligned} \tag{2}$$

The cost model $f$ predicts score $y$ of the tensor program $p$. The accuracy of the cost model $f$ is crucial in finding ideal optimization configuration.

### 3.2 Performance Model

The process of optimization using our design is outlined in Algorithm 1. The input is a set of to-be-optimized operators or subgraphs with different configurations. To implement our workflow, three functions are defined: GenerateHighSketch(), Sampling(), and EvolutionSearch(), as shown in Algorithm 1. GenerateHighSketch() takes $\phi_1$, $\sigma$, and $k$ as input and returns the high-level generation sketch $GS_1$ as output. Sampling() takes $GS_1$, $\phi_2$, $\sigma$, and $k$ as input and returns the low-level annotation samples $GS_2$ as output.

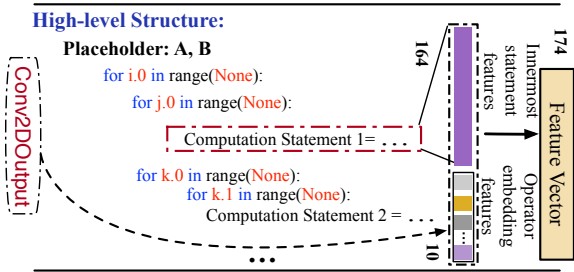

Figure 2: Hierarchical features of Conv2D with a full tensor program representation in the search space.

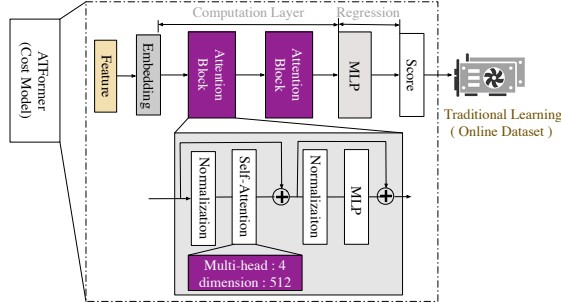

Figure 3: The performance model's architecture includes two attention blocks that extract coarse and fine-grained features of the tensor program, as well as a lightweight MLP layer for directly predicting the score.

EvolutionSearch() takes the high-level generation sketch $GS_1$ and the low-level annotation samples $GS_2$ as input and returns a group of tensor candidates for the cost model training. Next, an evolutionary search strategy is used along with a learned cost model to fine-tune the performance of the generated tensor programs. By iteratively mutating high-quality tensor programs, it can generate new programs with potentially higher quality. After a number of measurement trials, the best tensor program configurations can be identified.

**Hierarchical Feature Generation.** The input of ATFormer is a series of mix-grained feature vectors extracted from $p_\sigma$, where $p_\sigma$ is the full tensor program to implement operator $\sigma$. Each vector represents a single computation statement within $p_\sigma$. These mix-grained feature vectors are composed of two important components: *(i) Coarse-Grained operator embedding features* that capture the high-level structure of the operator $\sigma$ and *(ii) Fine-Grained statement features* that capture the low-level details of each statement within program $p_\sigma$. Each operator in the subgraph $S$ can be classified into a few categories, and we represent each operator with a one-hot embedding feature vector that covers all possible operator types. In practice, we use feature vectors of length 10 for the operator embedding and length 164 for the statement features, consistent with the approach used in Ansor (Zheng et al., 2020). The prediction score for a subgraph is computed as the sum of the prediction scores for each innermost non-loop statement within the loop nests of the full tensor program. More details can be found in Figure 2.

**Model Architecture** Our proposed ATFormer model consists of three layers: (i) a kernel embedding layer, which extracts a compact feature representation; (ii) a computation processing layer, which captures essential information from the innermost non-loop computation statements in the

neighborhood; and (iii) a simple regression layer for making the final prediction. ATFormer can be easily integrated into existing search algorithms and consistently improve the efficiency of auto-tuning. We believe that the simplicity of our method will attract more research attention to the field of tensor operator optimization, further enhancing training and inference efficiency. The feature processing of computation and regression in ATFormer is illustrated in Figure 3. The kernel embedding layer is composed of two fully connected layers with ReLU activation. The function of the kernel embedding layer is to project the features from low dimension space to a new embedding space for similarity measurement. Starting from the batched tensor programs $\mathcal{I} \in \mathbb{R}^{L \times D_{in}}$ representing a specific type of operator $\sigma$, where $L$ is the accumulated number of the feature statements within $\mathcal{I}$. A kernel embedding layer then generates a set of feature statements $\mathcal{E} \in \mathbb{R}^{L \times D_{out}}$ in embedding space. Typically, we use $D_{out} = 512$. The value $L$ is determined by the parameters of high-level structures $\phi_1$ and the low-level details sampling $\phi_2$ for each subgraph $S$.

As for the computation layer, a set of feature statements $\mathcal{E} \in \mathbb{R}^{L \times D_{out}}$ should be split into $M$ stacks of feature statements $\mathcal{Z} \in \mathbb{R}^{M \times N \times D_{out}}$ firstly. Each stack contains $N$ feature statements of innermost non-loop computation within a full tensor program $p$. We adopt the self-attention mechanism for feature statements aggregation. With the parameter tensors written as $\boldsymbol{W}^Q, \boldsymbol{W}^K, \boldsymbol{W}^V$, a full tensor program with a set of innermost non-loop feature statements $\boldsymbol{Z}$ is first encoded into query $\boldsymbol{Q}$, key $\boldsymbol{K}$, and value $\boldsymbol{V}$ by three identical linear transformations: $\boldsymbol{Q}, \boldsymbol{K}, \boldsymbol{V} = \boldsymbol{Z}^\top \boldsymbol{W}$. Then it will be further calculated by the self-attention

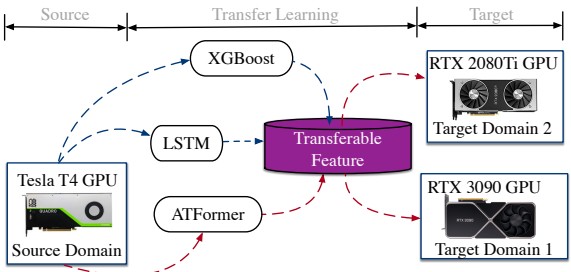

Figure 4: Transfer learning among different platforms.

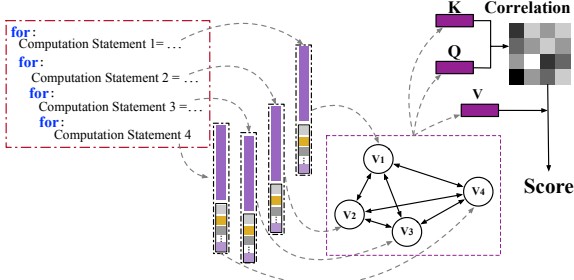

Figure 5: Self-attention between statement features.

layer as:

$$\text{Attention}(\boldsymbol{Q}, \boldsymbol{K}, \boldsymbol{V}) = \text{Softmax}\left(\frac{\boldsymbol{Q}^{\top}\boldsymbol{K}}{\sqrt{d_k}}\right)\boldsymbol{V}. \tag{3}$$

The final prediction of these $M$ tensor programs is computed by a regression layer with a dimension from 512 to 1. The predicted score is $\boldsymbol{y} \in \mathbb{R}^{M \times 1}$.

**Loss Function** The model ranks the performance of potential candidates in a large search space. Therefore, the model can be trained with ranking losses or regression losses to predict relative or absolute scores. To explore the loss function to train ATFormer, a common choice is to use the squared error function as a regressor which can mostly care about identifying the well-performing tensor programs. The loss function of the model $f$ on a full tensor program $p$ with throughput $h$ is $\text{MSELoss}(f, p, h) = (\sum_{s \in S(p)} \hat{f}(s) - y)^2$, where $S(p)$ is the set of innermost non-loop computation statements in tensor program $p$. We train ATFormer as the performance model $f$. However, we only care about the relative order of tensor program runtime rather than their absolute values during the compilation. We instead use the following RankLoss (Cao et al., 2007) to rank the performance of candidates in the large design space. This can fully exploit the optimal candidates to reduce the impact of the search algorithm on final prediction results. The loss function is defined as follows:

$$\text{RankLoss} = \sum_{s(i), s(j) \in S(p)} \log(1 + e^{f(i,j)}); \tag{4}$$

$$f(i, j) = -\text{sign}(y_i - y_j)(\hat{f}(s_i) - \hat{f}(s_j)). \tag{5}$$

We can use the prediction $\hat{f}(x)$ to select the top-performing implementations of a full tensor program $p$. The computation graph $G$ is trained for tensor programs extracted from all subgraphs. The throughput of all tensor programs is normalized to be in the range of $[0, 1]$.

### 3.3 Transfer Learning

The trade-off between search time and performance improvement is interesting to explore and exploit, as long search times may not always be acceptable. Our current focus is on developing a cost model for optimizing tensor operators on a specific hardware platform. However, in practical settings, we require a cost model that can be used across various hardware platforms. This would allow us to reuse a single-cost model for multiple platforms by providing it with new online data during auto-tuning. To achieve this, we pre-train the cost model with an offline static dataset and exploit transferable features that are invariant to both source and target domains to speed up the optimization process, as depicted in Figure 4. The use of transferable features greatly contributes to the success of transfer learning, as different designs may have varying degrees of invariance. By training the cost model offline using a dataset, we can significantly reduce the frequency of on-device measurements and use the pre-trained parameters as a starting point for new search tasks via transfer learning.

## 4 Experiments

### 4.1 End-to-End Execution Evaluations

**Workloads.** We evaluate the performance of ATFormer on various DNNs, including small and large-scale models. For small-scale models, we use AlexNet, VGG-16, MobileNet-V2, ResNet-18/50 and Bert-Tiny to evaluate the design. As for the large-scale models, we use BERT and GPT-3 models, specifically $\text{BERT}_{base}$, $\text{BERT}_{large}$, $\text{GPT-2}_{large}$ and $\text{GPT-3}_{350M}$. We report the the end-to-end inference latency with batch size 1 on RTX 2080Ti.

**Baselines and Settings.** For statistic model, we use XGBoost as a baseline which has proven to be a state-of-the-art feature-based model in auto-tuning framework (Zheng et al., 2020). For DNN-based

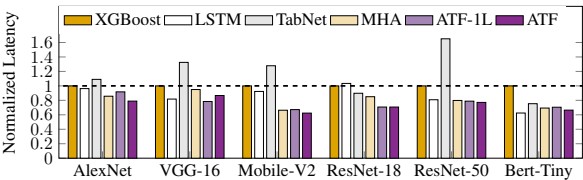

Figure 6: End-to-end performance comparison of cost models across DNNs and normalized by the XGBoost.

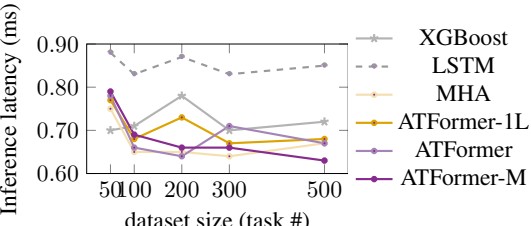

Figure 7: Transfer learning performance on TenSet.

learning, we use LSTM with eight heads and 1024 hidden dimensions, and TabNet is implemented in TenSet as another baseline. Note that the search algorithm uses the default configurations, and the search terminates when it runs out of allowed measurement trials. We keep the rest of the factors the same for a fair comparison.

**Main Results.** Figure 6 shows the final optimized *total latency* results on the RTX2080Ti GPU. Overall, the ATFormer-series model performs the best in all cases. Compared with the tree-based model XGBoost, ATFormer outperforms them in all cases with $1.15 - 1.61\times$ speedup. Compared with the DNN-based model TabNet, ATFormer outperforms them in all cases with $1.14 - 2.14\times$ speedup. Compared with LSTM, ATFormer performs equally the best and achieves $0.96 - 1.48\times$ speedup. Although LSTM surpasses ATFormer a little in finding the best configuration on Bert-Tiny and VGG-16, the amount of computation that can be parallelized in ATFormer leads to a shorter used time. Overall, the experiment results from the GeoMean verify the effectiveness of the attention-based modules over the tree- and DNN-based performance models.

### 4.2 Transfer Learning Evaluations

As mentioned in Section 3.3, we use RTX 2080Ti and 3090 GPUs as different platforms to verify our design by two typical metrics: *i) Fix the measurement trails and compare the total latency* and *ii) Fix a converged latency, and then compare the search time to reach it*. To explore transferable features and fast adaptation of auto-tuning between different hardware platforms, ATFormer is pre-trained with a number of samples from TenSet and then fine-tuned using online datasets on different platforms. Therefore, we divide our experiment settings into "traditional learning" and "transfer learning" parts.

**Traditional Learning.** In Table 1, ATFormer achieves the best total latency on RTX 2080Ti, and it performs almost equally best with ATFormer-1L about *total latency* with a fixed measurement trail

on 3090 GPU. The results show that self-attention based models perform best in the final performance compared with the tree-based and DNN-based cost models on different types of GPUs.

**Transfer Learning.** In Table 1, experiment results on RTX 2080Ti and 3090 show that the pre-trained parameters make the search convergence much faster. With the increasing number of training tasks in the offline dataset from 50 to 500, the learning ability of cost models with self-attention blocks, including MHA, ATFormer-1L, and ATFormer-Mask, become more stable, and they can adapt to the new tasks via transfer learning. ATFormer-series model performs better than the statistic and DNN-based model XGBoost, LSTM in optimized *total latency* with the parameters trained from TenSet-100 to TenSet-500. All large-scale models are exported from Hugging Face, with a batch size of 1 and a maximum input sequence length of 512. As shown in Table 2, ATFormer achieves latency speedups of $1.39\times$, $1.11\times$, $1.10\times$, and $1.16\times$ on the 3090 GPU compared to PyTorch runtime. In terms of end-to-end tuning time, ATFormer achieves speedups of $4.97\times$, $5.10\times$, $5.69\times$, and $6.08\times$ compared to traditional learning.

The performance of our efficient transfer learning on NVIDIA RTX 3090 GPU can be found in Figure 7. As for the TenSet-50 datasets, curves start from different points at the beginning, and we can find that XGBoost performs best. It means that the transferable features in the ATFormer-series models are not fully exploited on the limited dataset (task#50) during the training. Obviously, the adaptation skills amplify rapidly with the increasing number of tasks on the offline dataset. From TenSet-100 to TenSet-500, we can find that ATFormer-series models show fast adaptation and generalization ability across hardware platforms and operators compared with XGBoost and LSTM models.

In Table 3, we make the traditional learning and

| cost model (ms/s) | XGBoost latency | time | LightGBM latency | time | LSTM latency | time | TabNet latency | time | MHA latency | time | ATFormer-1L latency | time | ATFormer latency | time | ATFormer-M latency | time |
|---|---|---|---|---|---|---|---|---|---|---|---|---|---|---|---|---|
| ResNet-18-2080Ti | 1.47 | 573 | 1.58 | 770 | 1.29 | 604 | 1.52 | 748 | 1.32 | 687 | 1.25 | 706 | **1.04** | 787 | 1.23 | 762 |
| RTX 2080Ti Transfer — TenSet-50 | 0.86 | 535 | 0.98 | 527 | 1.02 | 614 | 1.13 | 583 | 1.01 | 595 | 1.00 | 602 | 0.97 | 600 | 1.00 | 611 |
| TenSet-100 | 0.96 | 533 | 0.98 | 526 | 1.07 | 615 | 0.82 | 596 | 0.87 | 602 | 1.00 | 602 | 0.85 | 594 | 0.84 | 611 |
| TenSet-200 | 0.99 | 536 | 0.86 | 525 | 1.07 | 611 | 0.88 | 582 | 0.83 | 602 | 0.82 | 612 | 0.82 | 604 | 0.82 | 632 |
| TenSet-300 | 0.89 | 538 | 0.85 | 526 | 1.02 | 622 | 0.83 | 583 | 0.85 | 600 | 0.81 | 609 | 0.89 | 612 | 0.87 | 607 |
| TenSet-500 | 0.96 | 530 | 0.81 | 529 | 1.03 | 622 | 0.82 | 574 | 0.83 | 593 | 0.87 | 598 | 0.84 | 612 | 0.79 | 615 |
| ResNet-18-3090 | 1.07 | 589 | 1.11 | 676 | 1.24 | 762 | 1.64 | 741 | 1.11 | 658 | **0.97** | 661 | **1.02** | 677 | 3.01 | 665 |
| RTX 3090 Transfer — TenSet-50 | 0.70 | 537 | 0.74 | 524 | 0.88 | 593 | 0.75 | 581 | 0.75 | 610 | 0.77 | 605 | 0.78 | 599 | 0.79 | 604 |
| TenSet-100 | 0.71 | 540 | 0.73 | 526 | 0.83 | 599 | 0.67 | 620 | 0.65 | 607 | 0.68 | 601 | 0.66 | 606 | 0.69 | 614 |
| TenSet-200 | 0.78 | 534 | 0.68 | 526 | 0.87 | 582 | 0.70 | 589 | 0.65 | 612 | 0.73 | 599 | 0.64 | 596 | 0.66 | 611 |
| TenSet-300 | 0.70 | 536 | 0.68 | 531 | 0.83 | 616 | 0.66 | 585 | 0.64 | 617 | 0.67 | 595 | 0.71 | 607 | 0.66 | 613 |
| TenSet-500 | 0.72 | 535 | 0.67 | 540 | 0.85 | 618 | 0.69 | 587 | 0.67 | 591 | 0.68 | 581 | 0.67 | 607 | 0.63 | 609 |

Table 1: Transferable adaptation evaluation between different GPU platforms on ResNet-18.

| cost model performance (ms / s) | XGBoost latency | time | LSTM latency | time | MHA latency | time | ATFormer-1L latency | time | ATFormer latency | time | Speed up latency | time |
|---|---|---|---|---|---|---|---|---|---|---|---|---|
| $BERT_{base}$ Traditional Learning | 24.51 | 3028 | 32.89 | 3246 | 19.13 | 2890 | 18.77 | 2996 | 17.56 | 2874 | 1.39× | 4.97× |
| $BERT_{base}$ Transfer Learning | 23.82 | 654 | 33.35 | 880 | 19.98 | 602 | 19.51 | 648 | 18.72 | 578 | | |
| $BERT_{large}$ Traditional Learning | 51.63 | 5016 | 59.81 | 5540 | 53.21 | 5218 | 54.32 | 5312 | 46.54 | 5232 | 1.11× | 5.10× |
| $BERT_{large}$ Transfer Learning | 52.49 | 1098 | 60.33 | 1302 | 55.88 | 1084 | 56.58 | 1192 | 47.76 | 1026 | | |
| GPT-2$_{large}$ Traditional Learning | 489.12 | 6240 | 502.22 | 6531 | 467.22 | 6311 | 452.56 | 6380 | 445.52 | 6268 | 1.10× | 5.69× |
| GPT-2$_{large}$ Transfer Learning | 491.24 | 1392 | 503.52 | 1594 | 468.29 | 1375 | 454.18 | 1272 | 447.31 | 1102 | | |
| GPT-3$_{350M}$ Traditional Learning | 513.61 | 7789 | 542.23 | 8582 | 479.42 | 8082 | 468.59 | 7982 | 442.02 | 7891 | 1.16× | 6.08× |
| GPT-3$_{350M}$ Transfer Learning | 514.42 | 1857 | 543.59 | 2302 | 480.12 | 1890 | 470.52 | 1920 | 443.62 | 1296 | | |

Table 2: The performance of large-scale Transformer models on TenSet-500 with transfer learning.

| cost model performance (ms / s) | XGBoost latency | time | LSTM latency | time | MHA latency | time | ATFormer-1L latency | time | ATFormer latency | time | ATFormer-M latency | time |
|---|---|---|---|---|---|---|---|---|---|---|---|---|
| RTX 2080Ti Traditional Learning | 1.26 | 1026 | 1.02 | 1487 | 1.03 | 1172 | 1.20 | 1269 | 1.02 | 1382 | 1.71 | 1124 |
| RTX 2080Ti Transfer Learning | 1.23 | 281 | 1.05 | 348 | 0.99 | 261 | 1.15 | 264 | 0.99 | 271 | 0.93 | 266 |
| RTX 3090 Traditional Learning | 0.96 | 1004 | 1.03 | 1235 | 0.79 | 1125 | 0.87 | 1141 | 0.74 | 2054 | 0.94 | 2018 |
| RTX 3090 Transfer Learning | 0.98 | 287 | 1.02 | 270 | 0.77 | 261 | 0.83 | 269 | 0.76 | 267 | 0.65 | 264 |

Table 3: Pre-trained models on TenSet-500 via transfer learning with converged latency on GPU platforms.

| Methods | ResNet-18 (a) | (b) | (c) | (d) | (e) | (f) | MobileNet-V2 (a) | (b) | (c) | (d) | (e) | (f) | Bert-Tiny (a) | (b) | (c) | (d) | (e) | (f) |
|---|---|---|---|---|---|---|---|---|---|---|---|---|---|---|---|---|---|---|
| mask? | | | ✓ | ✓ | | | | | ✓ | ✓ | | | | | ✓ | ✓ | | |
| pre-trained? | | | | ✓ | ✓ | | | ✓ | | ✓ | ✓ | | | ✓ | | ✓ | ✓ | |
| RMSE Loss? | ✓ | | | | | | ✓ | | | | | | ✓ | | | | | |
| Rank Loss? | | ✓ | ✓ | ✓ | ✓ | ✓ | | ✓ | ✓ | ✓ | ✓ | ✓ | | ✓ | ✓ | ✓ | ✓ | ✓ |
| AutoTVM? | | | | | | ✓ | | | | | | ✓ | | | | | | ✓ |
| total latency (ms) | 1.42 | 1.04 | 1.23 | **0.81** | 0.83 | 1.92 | 0.53 | 0.51 | 0.76 | **0.39** | 0.40 | 1.29 | 4.18 | 3.41 | 3.97 | **2.32** | 2.46 | 5.07 |
| search time (s) | 781 | 787 | 762 | **620** | 611 | 3274 | 962 | 1000 | 958 | **617** | 604 | 2996 | 1127 | 1141 | 1150 | **818** | 816 | 3826 |

Table 4: Total latency and tuning time of different methods, using ResNet-18, MobileNet-V2 and Bert-Tiny networks for end-to-end evaluation. The relative gains obtain for batch size = 1 with 300 measurement trials.

transfer learning on different hardware platforms for ResNet-18 have an approximate converged latency. ATFormer reduces the *search time* by up to 5.1× while maintaining the same search quality on RTX 2080Ti. This is the best speedup compared with 3.6× by XGBoost, 4.2× by LSTM, 4.8× by ATFormer-1L, and 2.2× MHA, respectively. Under the same conditions, ATFormer also performs the best with reducing the *search time* by up to 7.7× on RTX 3090 compared with 3.4× by XGBoost, 4.5× by LSTM, 4.2× by ATFormer-1L, respectively. Traditional learning with a mask-guided training scheme degrades the performance on *total latency* and *search time*. However, transfer learning

with a mask-guided training scheme for ATFormer-Mask performs best in most cases. Comprehensive experiments show that it is not easy to make ATFormer-Mask have the approximate converged latency on RTX 2080Ti and 3090 compared with traditional learning and transfer learning. It means that ATFormer-Mask with pre-trained parameters has better task generation for tensor programs and achieves better performance during tuning. Transfer learning across different types of CPUs can be found in Appendix A.6.

Overall, ATFormer takes full advantage of transferable features learned from the source domain Tesla T4 GPU and transfers the knowledge to the

| architecture | n_head | hidden_dim | latency (ms) | search time (s) |
|---|---|---|---|---|
| | 2 | 512 | 3.71 | 652 |
| | 4 | 256 | 1.58 | 647 |
| MHA | **4** | **512** | **1.24** | **641** |
| | 4 | 1024 | 1.29 | 652 |
| | 6 | 768 | 1.48 | 658 |
| | 8 | 512 | 1.19 | 658 |
| ATFormer-1L | 4 | 512 | 1.25 | 706 |
| **ATFormer** | **4** | **512** | **1.04** | 777 |
| ATFormer-3L | 4 | 512 | 1.23 | 788 |

Table 5: Different architecture design about ATFormer.

different target domains RTX 2080Ti and RTX 3090 to accelerate the convergence speed with a fixed number of measurement trails. Fast convergence is desirable for many users of auto-tuning to have better control of the optimization cost and good performance. For instance, deployment engineers may want to obtain an optimized model as soon as possible or quickly get an upper-bound estimation of total inference latency in real-world production. They can use the cost model like ATFormer with strong generalization as decent pre-trained parameters to accelerate not only the convergence speed but also the total execution inference time. Finally, comprehensive experiments with pre-trained parameters on different sizes of the TenSet dataset show that ATFormer-series models enable fast adaptation in not only cross-operator but also cross-platform scenarios.

| Methods | ResNet-18 | | | | | |
|---|---|---|---|---|---|---|
| | (a) | (b) | (c) | (d) | (e) | (f) |
| Hierarchical features? | ✓ | | ✓ | | ✓ | |
| XGBoost? | ✓ | ✓ | | | | |
| LSTM? | | | ✓ | ✓ | | |
| ATFormer? | | | | | ✓ | ✓ |
| w/o Transfer total lantency (ms) | **1.47** | **1.63** | 1.29 | 1.58 | **1.04** | **1.18** |
| w/o Transfer search time (s) | 573 | 618 | 604 | 648 | 787 | 796 |
| w/ Transfer total latency (ms) | 0.96 | 0.98 | 1.03 | 1.12 | **0.84** | 0.91 |
| w/ Transfer search time (s) | 530 | 599 | 622 | 689 | 612 | 632 |

Table 6: Hierarchical features and performance model architecture improvements for end-to-end evaluation.

## 4.3 Ablation Study

Various designs are evaluated in this section. We report the performance about *total latency*, *search time* on ResNet-18 and MobileNet-V2 and *accuracy* on the static datasets.

**Loss Functions.** Table 4 shows two different loss functions in our experiments. Method (a) is ATFormer with Root Mean Square Error (RMSE) loss function while method (b) is with lambdaRank loss function. Compared with method (a) and method (b), we find that lambdaRank loss always outperforms RMSE in our design for different

workloads of DNNs. It shows that the goal of a decent cost model is to rank the performance of different tensor programs by relative scores in a given search space.

**Convergence Speed.** In Table 4, method (d) is the proposed ATFormer, which adapts the pre-trained parameters to the new task via transfer learning into method (c). Note that ATFormer with the pre-trained parameters minimizes the *total latency* of all subgraphs in three DNNs as much as possible and the *search time* as quickly as possible. The proposed ATFormer improves the *total latency* by $4.66\times$ speedup and convergence speed by $1.55\times$ speedup. Method (f) is the AutoTVM with lambdaRank loss function. The performance is inferior to the baseline configuration.

**Training Schemes.** In Table 4, method (c) incorporates the mask module into method (b) during traditional learning. Method (d) imports the mask module into method (e) during transfer learning, resulting in a notable increase in convergence speed. It's worth noting that adding a mask scheme during traditional learning is not very helpful and can even cause a decrease in the total latency. However, for transfer learning with pre-trained parameters, incorporating the mask module is crucial for achieving faster convergence speed. The introduced techniques do not require expensive training resources in terms of both time and computation power.

**Model Architectures.** Table 5 lists ATFormer with various architectures. To achieve high accuracy while minimizing the model parameters, we find that the self-attention block, which contains four heads with 512 hidden dimensions, performs the best on the total latency and search time. Note that ATFormer does *not* benefit from deeper encoder layers in the Transformer model. Thanks to its simple and efficient architecture, the inference latency of ATFormer is consistently lower than that of the DNNs it optimizes. Thus, we set the two encoder layers as the final decision. Table 6 shows the relationship between the hierarchical-level features and different architectures to affect *total latency* and *search time* on ResNet-18.

**Accuracy.** Table 7 presents the pairwise comparison accuracy of ATFormer and XGBoost on various scales of static datasets. The findings indicate that ATFormer outperforms XGBoost, demonstrating the highest measurement accuracy and providing optimal search quality during the tuning. We successfully conduct the training process on a server

| Architecture | XGBoost | ATFormer-1L | ATFormer | ATFormer-M |
|---|---|---|---|---|
| TenSet-50 | 91.31 | 85.98 | **93.48** | 93.28 |
| TenSet-300 | 92.24 | 90.41 | **93.82** | 93.33 |
| TenSet-500 | 93.08 | 91.98 | **94.06** | 93.71 |

Table 7: Accuracy of the cost models on TenSet.

| Cost Model | TenSet-50 | TenSet-100 | TenSet-200 | TenSet-300 | TenSet-500 |
|---|---|---|---|---|---|
| ATFormer-1L | 258 | 362 | 549 | 685 | 916 |
| ATFormer | 299 | 384 | 588 | 712 | 951 |
| ATFormer-M | 324 | 416 | 605 | 749 | 972 |

Table 8: The training time of the ATFormer series cost models during the offline optimization.

| Model | XGBoost | LSTM | TableNet | MHA | ATFormer-1L | ATFormer-2L |
|---|---|---|---|---|---|---|
| ResNet-18 | 3.04 | 3.35 | 2.88 | 2.79 | 2.51 | **2.39** |
| Bert-Tiny | 17.42 | 14.83 | 16.98 | 15.32 | 15.49 | **14.37** |

Table 9: Traditional learning with different cost models for batch size 8 on the NVIDIA RTX 3090 GPU.

equipped with an Intel Core i9-12900K CPU, a NVIDIA GeForce RTX 3090 GPU, and a 2TB hard disk. Table 8 presents the specific training times (in seconds) of the ATFormer series models on static datasets. Note that our approach is also suitable for scenarios involving large batch sizes. Table 9 lists experimental results using batch size 8 on the NVIDIA 3090 GPU via traditional learning.

## 5 Conclusion

This paper introduces ATFormer, a novel and effective design for optimizing tensor programs. ATFormer employs hierarchical features with varying levels of granularity to model the end-to-end compilation. Moreover, self-attention blocks are utilized to explore global dependencies of a complete tensor program for high-quality evaluation. Through transfer learning, ATFormer achieves faster-converged latency and superior transferability across different hardware platforms, outperforming previous state-of-the-art benchmarks.

**Limitations.** We plan to do the transfer learning from GPUs to CPUs and explore the potential of combining with post-training quantization or pruning to efficiently deploy models. Additionally, we will explore more universal and efficient methods for optimizing tensor programs with ATFormer. This includes leveraging hardware features to optimize performance on domain-specific accelerators, such as NVIDIA's Tensor Cores.

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

# A Appendix

## A.1 Feature Extraction Details

The feature before ATFormer training can be represented as two different granularities: coarse-grained and fine-grained levels. The coarse-grained level feature can describe each search task in the computation graph. It has **10** elements with the one-hot encoding pattern. In our specific code implementation, the coarse-grained vector contains these operators:"max", "min", "add", "Conv2dOutput", "Conv2d_winograd", "DepthwiseConv2d", "dense", "softmax", "compute(b, i, j)". The "max" and "min" can represent some activation functions in deep learning. "dense" means the fully connected layer in computation graph and "compute(b, i, j)" is a very important function to implement each tensor operation in deep learning. If the intermediate representation about some operators are fused into the same "compute(b, i, j)" primitive, it means these operators are fused together and can run very efficiently on the specific hardware platforms. As for the fine-grained vector, the length of it including all the listed features for one statement is **164**. We use the same set of features for both Turing 2080Ti and Ampere 3090 GPUs. It can be summarized as follows:

- **Number of float operations:** The number of addition, subtraction, division, modulo operation, less-than, greater-than, intrinsic math function such as exp, sqrt.

- **Number of integer operations:** Similar to the `number of float operations`, but for the operations with integer operations.

- **Vectorization related features:** The number of the innermost vectorized loop statements in a full tensor program.

- **Unrolling related features:** The number of the innermost unrolling loop statements in a full tensor program.

- **Parallelization related feature:** The number of the innermost parallelization loop statements in a full tensor program.

- **GPU thread binding related features:** The lengths of blockIdx.x, blockIdx.y, blockIdx.z, threadIdx.x, threadIdx.y, threadIdx.z and virtual threads which can avoid bank conflict problem in shared memory.

- **Arithmetic intensity curve:** We only sample 10 points from a curve which is defined as $\frac{FLOPs}{Bytes}$ which is similar to the `roof-line` model used in computer architecture. It can help us to recognize the type of the search task or operator in computation graph such as compute-intensive or memory-intensive operator on a specific hardware platform.

- **Buffer Access Feature:** We perform feature extraction for at most five buffers. It includes "Access type", "Bytes", "Unique bytes", "Lines", "Unique Lines", "Reuse type", "Reuse distance", "Reuse counter", "Stride", "Accessed bytes divided by reuse".

- **Allocation related features:** The size of the allocated buffer for the output results of each statement in a full tensor program.

With the combination of coarse-grained and fine-grained feature vectors, we can construct them into a hierarchical feature vector to take full advantage of each statements in a full tensor program.

## A.2 Implementation Details

ATFormer is implemented on the top of Ansor and evaluated from two aspects: end-to-end search efficiency and quality, as well as performance portability. We compare ATFormer against the state-of-the-art methods, including both the statistic and DNN-based cost models. The items labeled with XGBoost represent the Ansor default configuration. We also provide a detailed ablation study of the model architecture, accuracy, loss function, convergence speed, and training scheme, with insights and qualitative results. The generated tensor programs are evaluated on two different GPU architectures: Turing RTX 2080Ti and Ampere RTX 3090, with float32 data types used for all evaluations. We train the cost model using the Adam optimizer for 50 epochs, with a starting learning rate of $7e^{-4}$ that decays to $1e^{-6}$, and a training batch size set to 512. We use TVM v0.8dev in TenSet (Zheng et al., 2021), LLVM 11.0, and CUDA 11.0 for compilation, while XGBoost 1.5.0 and PyTorch 1.7.1 are used for training models. The use of a "mask" is a widely adopted technique for training transformers. In Figure 5, each tensor program is transformed into a sequence of vectors, with each vector representing a tensor computation statement. During training, all sequences are of the same length, and any shorter sequences are

padded with zeros at the end. The padded items are masked out and excluded from the loss computation. Our ablative models, including MHA, ATFormer-1L, ATFormer, and ATFormer-M, were also experimented with. MHA is the basic Multi-Head Attention layer, ATFormer-1L only has one encoder layer, ATFormer has two encoder layers, and ATFormer-M uses the "mask" scheme during training.

### A.3 Dataset Details

We evaluated our design using TenSet, a large-scale and challenging dataset for search-based tensor compilers. TenSet comprises 52 million performance records of tensor programs obtained from real measurements on different hardware platforms. Various randomly generated tensor programs for popular workloads are compiled via the TVM compiler and executed on the target hardware platforms. To ensure the inclusion of diverse workloads essential for generalization ability, we collected tensor programs from 120 networks with 13,848 tasks on the NVIDIA Tesla T4 GPU. This dataset serves as a series of static offline datasets.

### A.4 Benchmark Details

We evaluate the performance of generated programs by ATFormer on two levels: end-to-end network evaluations and performance portability via transfer learning. For each level of evaluation, we compare ATFormer against the state-of-the-art methods, including the statistic models:

- XGBoost (Chen and Guestrin, 2016b)

- LightGBM (Ke et al., 2017)

and DNN-based models:

- LSTM (Hochreiter and Schmidhuber, 1997)

- Multi-Head Attention (Vaswani et al., 2017)

- TabNet (Arik and Pfister, 2021)

The generated tensor programs are benchmarked on two different architecture GPU platforms:

- NVIDIA 2080Ti GPU with Turing architecture (Jia et al., 2019)

- NVIDIA 3090 GPU with Ampere architecture (Choquette et al., 2021)

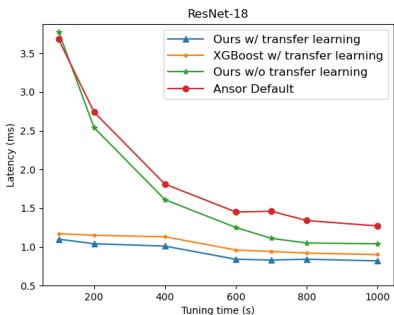

Figure 8: Convergence analysis on ResNet-18.

We use float32 as the data type for all evaluations. We train our model with the Adam optimizer for 50 epochs with a starting learning rate of $7e^{-4}$, the learning rate decays to $1e^{-6}$, and the training batch size is set to 512. We use TVM v0.8dev in TenSet, LLVM 11.0 and CUDA 11.0 for compilation. Meanwhile, we use XGBoost 1.5.0 and PyTorch 1.7.1 for training models.

To explore transferable features and fast adaptation of ATFormer between different hardware platforms, ATFormer is pre-trained using offline learning with a number of samples from TenSet, and then fine-tuned using online learning on different platforms. For the offline learning, we randomly sample 50, 100, 200, 300, 500 search tasks from TenSet NVIDIA Tesla T4 GPU.

We train 40 models including XGBoost, Light-GBM, LSTM, TabNet, Multi-head attention, ATFormer-1L, ATFormer, ATFormer-Mask for all of experiment evaluation in this paper. Due to the limitation of maximum file size (100MB) in supplementary material, we release the pre-trained model offline learning by Tenset-500 for AFTormer-1L, ATFormer, ATFormer-Mask, Multi-head attention and TabNet. All of the pre-trained models for XG-Boost. And we release running scripts in the supplementary material to reproduce the results in Section 5 Table 1. More details about the hyperparameters of each cost model in our experiments can be found in Table 12, Table 13, Table 14, Table 15, Table 16, Table 17, and Table 18.

### A.5 Convergence Analysis.

In Figure 8, we present the tuning trials-latency curves that illustrate various stages of auto-tuning with different configurations on ResNet-18. We performed four types of experiments on ResNet using two settings: with transfer learning and without transfer learning. The blue line indicates ATFormer with transfer learning to expedite the tuning pro-

| cost model performance (ms / s) | | XGBoost | | LSTM | | MHA | | ATFormer-1L | | ATFormer | |
|---|---|---|---|---|---|---|---|---|---|---|---|
| | | latency | time | latency | time | latency | time | latency | time | latency | time |
| ResNet-18 | Traditional Learning | 6.13 | 658 | 6.16 | 731 | 6.12 | 642 | 6.22 | 633 | 6.15 | 661 |
| | Transfer Learning | 6.16 | 334 | 6.25 | 451 | 6.19 | 346 | 6.29 | 419 | 6.18 | 304 |
| ResNet-50 | Traditional Learning | 19.59 | 652 | 21.23 | 697 | 17.50 | 630 | 17.52 | 614 | 16.90 | 643 |
| | Transfer Learning | 20.01 | 342 | 21.99 | 461 | 18.11 | 338 | 17.91 | 362 | 17.02 | 318 |
| VGG-16 | Traditional Learning | 36.92 | 891 | 39.85 | 1004 | 35.69 | 839 | 34.51 | 826 | 30.01 | 840 |
| | Transfer Learning | 37.51 | 395 | 40.17 | 422 | 36.79 | 326 | 34.87 | 318 | 34.88 | 270 |
| BERT-Tiny | Traditional Learning | 17.98 | 1012 | 19.22 | 1433 | 17.55 | 1126 | 16.09 | 1168 | 15.11 | 1232 |
| | Transfer Learning | 18.05 | 396 | 19.57 | 498 | 17.91 | 401 | 16.41 | 416 | 15.16 | 388 |

Table 10: Pre-trained models with the converged latency on the CPU platforms.

| cost model performance (ms / s) | | XGBoost | | LSTM | | MHA | | ATFormer-1L | | ATFormer | |
|---|---|---|---|---|---|---|---|---|---|---|---|
| | | latency | time | latency | time | latency | time | latency | time | latency | time |
| ResNet-18 | Traditional Learning | 5.28 | 634s | 5.91 | 702 | 5.17 | 611 | 5.32 | 602 | 4.75 | 628 |
| | Transfer Learning | 5.21 | 314 | 5.88 | 432 | 5.16 | 326 | 5.19 | 384 | 4.74 | 254 |
| ResNet-50 | Traditional Learning | 16.42 | 621 | 18.23 | 632 | 13.51 | 608 | 12.51 | 584 | 11.62 | 602 |
| | Transfer Learning | 20.01 | 342 | 21.99 | 461 | 18.11 | 338 | 17.91 | 362 | 17.02 | 318 |
| VGG-16 | Traditional Learning | 29.52 | 845 | 31.54 | 967 | 28.55 | 799 | 28.71 | 796 | 25.49 | 812 |
| | Transfer Learning | 29.41 | 352 | 31.47 | 378 | 28.46 | 299 | 28.69 | 278 | 25.46 | 216 |
| BERT-Tiny | Traditional Learning | 13.88 | 862 | 15.22 | 1138 | 13.55 | 986 | 14.41 | 942 | 11.55 | 998 |
| | Transfer Learning | 13.76 | 339 | 15.47 | 438 | 13.91 | 345 | 14.39 | 377 | 11.58 | 320 |

Table 11: Pre-trained models with the converged latency on the Tensor Cores.

cess. We observe that the converged latency is the best among the four configurations. The orange line represents the same tuning process with the XGBoost cost model, and we note that the converged latency is inferior to the one with ATFormer. The green line shows ATFormer without transfer learning, and we can observe that the convergence speed is exceptionally fast. The red line represents the Ansor optimization, and we observe that the convergence speed and the final converged latency are both less than the ones achieved by the green line with ATFormer. Therefore, we can infer that ATFormer can expedite the tuning process compared to traditional learning methods through transfer learning and outperforms the state-of-the-art tensor compiler Ansor.

The main components in ATFormer model architecture can be categorized into three layers:

- **Kernel embedding layer:** The function of kernel embedding layer is to change the $164 + 10$ dimensions into $512$ dimensions.

- **Computation layer:** The function of computation layer is to obtain the relationship between each innermost non-loop statement in loop nests of a full tensor program.

- **Regression layer:** The function of regression layer is to project the final prediction about each innermost non-loop statement in an one

dimension scalar.

| Model Name | Parameter | Value |
|---|---|---|
| XGBoost | max_depth | 6 |
| | gamma | 0.003 |
| | min_child_weight | 2 |
| | eta | 0.2 |

Table 12: Hyperparameters of XGBoost.

| Model Name | Parameter | Value |
|---|---|---|
| LightGBM | num_leaves | 72 |
| | boosting_type | gbdt |
| | lr | 0.16 |
| | bagging_freq | 4 |
| | min_sum_leaf | 4 |
| | fraction_f | 0.84 |
| | fraction_b | 0.89 |

Table 13: Hyperparameters of LightGBM.

## A.6 Other Platforms: Intel CPUs

We use the dataset from Intel Platinum-8272 to verify transferability on Intel E5-2698 CPU with a fixed converged latency (6.13ms) by the same measurement trials for ResNet-18. More details can be found in Table 10. Therefore, ATFormer also works well for CPU with lots of different DNN

| Model Name | Parameter | Value |
|---|---|---|
| TabNet | in_dim | 164 + 10 |
| | hidden | 256 |
| | out_dim | 1 |
| | n_d | 1 |
| | n_a | 8 |
| | n_steps | 3 |
| | gamma | 1.3 |
| | lr | $7e^{-4}$ |

Table 14: Hyperparameters of TabNet.

| Model Name | Parameter | Value |
|---|---|---|
| LSTM | in_dim | 164 + 10 |
| | hidden_dim | 1024 |
| | out_dim | 1 |

Table 15: Hyperparameters of LSTM.

| Model Name | Parameter | Value |
|---|---|---|
| MHA | in_dim | 164 + 10 |
| | num_heads | 4 |
| | hidden_dim | 512 |
| | out_dim | 1 |

Table 16: Hyperparameters of Multi-head Attention.

| Model Name | Parameter | Value |
|---|---|---|
| ATFormer | in_dim | 164 + 10 |
| | num_heads | 4 |
| | hidden_dim | 512 |
| | num_layers | 2 |
| | out_dim | 1 |
| | padding_mask | False |

Table 17: Hyperparameters of ATFormer.

| Model Name | Parameter | Value |
|---|---|---|
| ATFormer-M | in_dim | 164 + 10 |
| | num_heads | 4 |
| | hidden_dim | 512 |
| | num_layers | 2 |
| | out_dim | 1 |
| | padding_mask | True |

Table 18: Hyperparameters of ATFormer.

benchmarks including ResNet-50, VGG-16, BERT-Tiny with batch size 1. As for the ResNet-18, we fix the converged latency to 19.59ms, the traditional learning will cost 658s to search the optimal configuration with XGBoost performance model. But the ATFormer can search the optimal implementation of ResNet-50 with 643s by the same measurement trials under the 16.90ms converged latency. We can get the same conclusions from the VGG-16 and BERT-Tiny neural networks.

## A.7 Performance on Tensor Cores

The recent advancements of GPU hardware technology have resulted in a significant increase in computing power, particularly with the introduction of the Tensor Cores on NVIDIA GPUs. Unlike the scalar-to-scalar primitives found in CPUs or the general CUDA Cores in GPUs, Tensor Cores provide specialized tensor computation capacities, which can deliver over $10\times$ higher throughput. Notably, the initial version of Tensor Core is designed for handling the GEMM with half-precision input and full-precision output. Recently, new features supporting different datatypes such as `int8`, `int4` and `int1` input variables have been introduced in the latest architecture (Truing and Ampere). The collection process takes 5 days with a server equipped with an Intel Core i9-12900K CPU and NVIDIA GeForce RTX 3090 GPU. The sampling selection process for the operator is conducted in a manner similar to that on the GPU's CUDA cores. We use the floating point 16 (`fp16`) as the experiemnt datatype and additional experimental results on transfer learning are presented in Table 11.