# OpenReview forum: "ATFormer: A Learned Performance Model with Transfer Learning Across Devices for Deep Learning Tensor Programs"
_EMNLP/2023/Conference — EMNLP 2023 Main_

### Official Review · Reviewer_fHck · 2023-08-03

**Soundness:** 3

**Excitement:**

4: Strong: This paper deepens the understanding of some phenomenon or lowers the barriers to an existing research direction.

**Paper Topic And Main Contributions:**

This paper provides a comprehensive analysis of auto-tuning challenges in code generation. Instead of relying on a statistical model like XGBoost, the proposed techniques leverage an attention-block-based model to capture the implicit long-context relationships and representations in tensor programs. Compared to other DNN-based algorithms, such as LSTM, the attention mechanism executes in parallel and requires less training time.

Moreover, the paper showcases the applicability of cross-device transfer learning, effectively transferring learned features from offline static datasets on one hardware platform to another new platform. This approach enables users to expedite the optimization/searching stage by reusing a cost model.

The study includes a detailed comparison experiment involving existing algorithms (XGBoost, LSTM, and MHA) on two NVIDIA GPU architectures: Turing and Ampere. These experiments encompass widely used classical CV and NLP models like VGG, ResNet, BERT, and GPT. Additionally, the paper conducts a transfer learning evaluation to gauge the performance of the transferred features during tuning for tensor program generation tasks.

**Questions For The Authors:**

1.	In Figure 3, why are the outputs of self-attention and embedding concatenated again? What is the reason for including an embedding layer in every attention block? Additionally, could you confirm whether the attention block utilizes pre- and post-layer norms?
2.	Please provide the latency values for BERT-large and GPT models in Figure 6? It seems that ATF offers little improvement compared to MHA in some cases.
3.	It seems that ATFormer with 2 attention blocks performs worse than ATFormer-1L on the dataset with a size of 300 points in Figure-7. Can you provide insights into the reasons behind this performance difference?
4.	In Table-2, it appears that ATF reduces search time by utilizing transfer learning, but this comes at the cost of increased kernel execution latency. Does this ultimately lead to a worse overall result?

**Reasons To Accept:**

1)Extensive experiments. The results can be useful.
2) An interesting approach. The proposed approach uses two parts of feature information to capture high-level and low-level representations separately: coarse-grained operator embedding and fine-grained statement.

**Reasons To Reject:**

The code generation templates are overly simple. The final prediction/tuning objective only considers the innermost non-loop statements. There is no optimization like loop tiling in the code generation task, which is shown to be important for code optimization of DNN workloads.

**Reproducibility:**

4: Could mostly reproduce the results, but there may be some variation because of sample variance or minor variations in their interpretation of the protocol or method.

**Reviewer Confidence:**

2: Willing to defend my evaluation, but it is fairly likely that I missed some details, didn't understand some central points, or can't be sure about the novelty of the work.

---

> ### Author Rebuttal · Authors · 2023-08-29
>
> We sincerely appreciate your valuable comments on our work. To address your concerns, our responses are demonstrated below:
>
> **Q1:** The code generation templates are overly simple. The final prediction/tuning objective only considers the innermost non-loop statements. There is no optimization like loop tiling in the code generation task, which is shown to be important for code optimization of DNN workloads.
>
> **A1:** Thanks for the comments. Our method does not solely optimize the innermost statements but rather focuses on optimizing the full tensor program, including conventional loop-tiling, reorder,unrolling and so on. In Section 3.1 of the problem formulation, we define a hierarchical search space consisting of two important components: high-level structure and low-level details. The high-level structure is responsible for making structural changes to the entire for-loop nest, including loop tiling, reordering, and unrolling. On the GPU, we utilize the "SSSRRSRS" template to generate the corresponding high-level structure. Here, "S" refers to one tiling level of space loops, and "R" represents one tiling level of reduction loops. The loops in the first three space tilings are bound to BlockIdx, virtual thread (for reducing bank conflicts), and ThreadIdx on the GPU, respectively. Different transformations in the outermost for-loops will result in different features extracted from the innermost assignment statement, leading to varying throughput predictions and learning processes in the cost model. Overall, our method considers various optimizations of the outermost loops during the optimization process. By extracting features from the innermost non-loop statements, we train a learnable cost model on the GPU to automatically optimize the full tensor program. We train the cost model and mainly target the data parallel tensor programs which consist outermost loop-nest and some innermost assignment statement. We use the ATFormer to predict the score of each non-loop innermost statement. An example are shown as follows:
>
> ```python
> for i in range(10):
>   for j in range(10):
>       B[i][j] = A[i] * 2                          # Statement B
> for i in range(10):
>   	C[i] = B[i][i] - 3                          # Statement C
> ```
>
> In this example, there are two non-loop innermost statement, B and C. We estimate the cost of these tensor programs. The total cost of it is the sum of cost B and cost C. We perform static analysis to extract the features for every non-loop innermost statement and make prediction for them. These predictions, along with features extracted from the entire program, such as the number of *used cached lines*, *used memory*, *reuse distance*, and *arithmetic intensity curves*  are used to estimate the cost of the entire program. In the following code snippet, we utilize an example of matrix multiplication and ReLU function to elucidate the high-level structure and low-level details as mentioned before.
> ```python
> for i in range(512):
>   for j in range(512):
>     for k in range(1024):
>       C[i, j] += A[i, k] * B[k, j]
> for i in range(512):
>   for j in range(512):
>     D[i, j] = max(C[i, j], 0)
> ```
> One of the possible high-level structure transformation is shown as below:
> ```python
> for i.0 in range(Tile_I0):
>   for j.0 in range(Tile_J0):
>     for i.1 in range(Tile_I1):
>       for j.1 in range(Tile_J1):
>         for k.0 in range(Tile_K0):
>           for i.2 in range(Tile_I2):
>             for j.2 in range(Tile_J2):
>               for k.1 in range(Tile_I1):
>                 for i.3 in range(Tile_I3):
>                   for j.3 in range(Tile_J3):
>                     C[...] += A[...] * B[...]
>         for i.4 in range(Tile_I2 * Tile_I3):
>           for j.4 in range(Tile_J2 * Tile_J3):
>             D[...] += max(C[...], 0.0)
> ```
> The following two code snippets demonstrate the low-level details that can be generated based on the aforementioned high-level structure in the hierarchical search space.
> ```python
> parallel i.0@j.0@i.1@j.1 in range(256):
>   for k.0 in range(32):
>     for i.2 in range(16):
>       unroll k.1 in range(16):
>         unroll i.3 in range(4):
>           vectorize j.3 in range(16):
>             C[...] += A[...] * B[...]
>   for i.4 in range(64):
>     vectorize j.4 in range(16):
>       D[...] = max(C[...], 0.0)
> ```
> ```python
> parallel i.2 in range(16):
>   for j.2 in range(128):
>     for k.1 in range(512):
>       for i.3 in range(32):
>         vectorize j.3 in range(4):
>           C[...] += A[...] * B[...]
> parallel i.4 in range(512):
>   for j.4 in range(512):
>     D[...] = max(C[...], 0.0)
> ```
> By examining the provided code snippets, it becomes evident that throughout the entire process of training and prediction in the cost model, there are considerations for common optimizations such as tiling that affect the outermost for-loops. To provide a more comprehensive understanding of our design and optimization methodology, we will showcase detailed code examples in the appendix. These examples will illustrate the process and incorporate common optimizations to enhance readers' comprehension.
>
> **Q2:** In Figure 3, why are the outputs of self-attention and embedding concatenated again? What is the reason for including an embedding layer in every attention block? Additionally, could you confirm whether the attention block utilizes pre- and post-layer norms?
>
> **A2:** Thanks for the comments. In Figure 3, the term "embedding" in the attention block refers to a vector representation of the features in an embedding space. The embedding layer is positioned between the features extracted from assignment statements and the first attention block, as illustrated in the following diagram.
> ```
> Feature => Embedding Layer => Attention Block => Attention Block => MLP => Score
> ```
> The function of the embedding layer is to project the discrete features from low dimension space to a continuous embedding space for importance measurement.  This enables the cost model to process the feature and consider the importance of assignment statement in subsequent self-attention mechanisms. Therefore, the embedding layer appears only once in the ATFormer series models. The specific implementation of the embedding layer is described below:
> ```python
>       self.embedding_layer = torch.nn.Sequential(
>             torch.nn.Linear(in_dim, hidden_dim),
>             torch.nn.ReLU(),
>             torch.nn.Linear(hidden_dim, hidden_dim),
>             torch.nn.ReLU(),
>         )
> ```
> The embedding layer consists of two consecutive MLP layers followed by a non-linear activation function (ReLU). Therefore, the term "embedding" in the Figure 3 does not refer to the embedding layer. We apologize for the confusion caused by the figure and sincerely appreciate the reviewer for pointing out this concern. In the final revised version, we will re-adjust the content of Figure 3 to enhance clarity and facilitate a better understanding of our structure by the readers. In this case, we do not concatenate the output of the embedding layer with the output of self-attention. Instead, we add the output of the embedding layer with a normalization layer to the output of self-attention in the first attention block. In the second attention block, we first pass the output of the first attention block through a normalization layer. Then, we add it to its own self-attention and further pass it through another normalization layer. This addition follows the same principle as the residual addition in standard transformer encoder. Yes. in our ATFormer design, each attention block incorporates two normalization layers: one located before the self-attention mechanism and another positioned after it. The purpose of the normalization layer is to stabilize the input distribution of the forward data and ensure stable gradient propagation during training.
>
> **Q3:** Please provide the latency values for BERT-large and GPT models in Figure 6? It seems that ATFormer offers little improvement compared to MHA in some cases.
>
> **A3:** Figure 6 illustrates the end-to-end network optimization using the traditional learning. In comparison to MHA, ATFormer achieves acceleration factors of 1.09x, 1.11x, 1.06x, 1.21x, 1.04x, and 1.05x for AlexNet, VGG-16, MobileNet-V2, ResNet-18, ResNet-50, and Bert-Tiny, respectively. Regarding Bert-large and GPT-2-large models, ATFormer achieves improvements of **1.11x** and **1.13x**, respectively, compared to MHA.  For further optimization results, please refer to the table provided below:
> | Cost Model| Bert-large (ms) | GPT-2-large (ms) |
> | ----------- | --------------- | ---------------- |
> | XGBoost     | 34.56           | 279.11           |
> | LSTM        | 31.22           | 269.58           |
> | MHA         | 29.98           | 262.74           |
> | ATFormer-1L | 29.49           | 251.42           |
> | ATFormer-2L | 27.01           | 232.51           |
>
> **Q4:** It seems that ATFormer with 2 attention blocks performs worse than ATFormer-1L on the dataset with a size of 300 points in Figure-7. Can you provide insights into the reasons behind this performance difference?
>
> **A4:** Thanks for the comments. Figure 7 presents the transfer learning capabilities of different cost models on the TenSet-50/100/200/ 300/500 datasets. It is discernible that the augmentation in inference performance differs among the various cost models when confronted with a relatively constrained offline dataset (TenSet-50).  Subsequently, as the offline dataset's sample count increases, the transfer learning capabilities demonstrated by the distinct cost models progressively evolve.Particularly, when examining TenSet-300, it becomes evident that the transfer learning prowess of ATFormer-2L lags behind that of ATFormer-1L. The underlying cause lies in the fact that ATFormer-1L possesses a smaller parameter size in comparison to ATFormer-2L, resulting in the latter model experiencing underfitting after training on a constrained offline dataset. Nonetheless, as the number of training samples within the dataset grows, the learning capability of ATFormer-2L demonstrates continuous improvement, eventually surpassing ATFormer-1L in terms of optimization performance. Overall, ATFormer-M showcases the most stable performance among all the cost models during the transfer learning. It provides strong evidence for the superiority of the mask mechanism when training attention-based cost models during the process of operator tuning.
>
> **Q5:** In Table-2, it appears that ATFormer reduces search time by utilizing transfer learning, but this comes at the cost of increased kernel execution latency. Does this ultimately lead to a worse overall result?
>
> **A5:** Thanks for the comments. In Table-2, the term "transfer learning" refers to the training of cost models such as XGBoost, LSTM, MHA, ATFormer-1L, and ATFormer (2L) on the Tenset-500 offline dataset, followed by subsequent online updates of the pre-training. On the contrary, "traditional learning" indicates the direct training of these cost models from scratch using online datasets. It is worth noting that the number of trials required during the tuning (search) process differs between these two approaches. The table clearly demonstrates a significant reduction in the tuning time taken for the networks to converge, for example, from 3028 to 654 seconds as for BERT-base model. In the "Speedup" metric, it represents the acceleration achieved at the latency level when tuning a network end-to-end with the same trials, such as a 1.39x speedup.  The "time" metric indicates the substantial acceleration in achieving approximate inference latency with both traditional learning and transfer learning, for instance, a 4.97x acceleration. More details about these two metrics can be found in line 432-435. Therefore, using transfer learning only accelerates the convergence time during network tuning and does not increase the cost of kernel execution or lead to worse results.

---

### Official Review · Reviewer_y1Wr · 2023-08-03

**Soundness:** 4

**Excitement:**

4: Strong: This paper deepens the understanding of some phenomenon or lowers the barriers to an existing research direction.

**Paper Topic And Main Contributions:**

The training and inference efficiency of deep neural networks heavily relies on the performance of tensor operators on specific hardware platforms. Compilation-based optimization flows, including automatic tensor generation and parameter tuning, are necessary for efficient model deployment. This paper presents ATFormer, a simple yet efficient design with attention-inspired modules to accurately predict the performance of optimized operators by capturing global and long-range dependencies within a complete scheduling space. ATFormer enables rapid adaptation of performance tuning across various GPU platforms using pre-trained parameters on static datasets.

Compared with the state-of-the-art baselines, ATFormer can predict the optimal implementation of tensor operators to reduce latency time with minimal effort on modern DNN benchmarks.

**Questions For The Authors:**

1) What are the high-level structure and low-level structure in hierarchical feature generation?

2) In Figure-5, how to incorporate statement features into self-attention and how to update self-attention after scanning the full tensor program?

3) Why is the throughput metric used in the loss function although the experiment results show a latency improvement?

4) Table-4 should include one or two nlp models.

**Reasons To Accept:**

1) The work is well-written, and the sufficient experiment results demonstrate the soundness of the authors' design.

**Reasons To Reject:**

1) The problem chosen for this study is simple, with only the innermost nonloop statement to be tuned.

**Reproducibility:**

3: Could reproduce the results with some difficulty. The settings of parameters are underspecified or subjectively determined; the training/evaluation data are not widely available.

**Reviewer Confidence:**

2: Willing to defend my evaluation, but it is fairly likely that I missed some details, didn't understand some central points, or can't be sure about the novelty of the work.

---

> ### Author Rebuttal · Authors · 2023-08-29
>
> Thank you for your comments and appreciation of our work. To relieve some of your concerns, we have posted our responses below:
>
> **A1:** Thanks for the comments. Our method does not solely optimize the innermost statements but rather focuses on optimizing the full tensor program, including conventional loop-tiling, reorder,unrolling and so on. In Section 3.1 of the problem formulation, we define a hierarchical search space consisting of two important components: high-level structure and low-level details. The high-level structure is responsible for making structural changes to the entire for-loop nest, including loop tiling, reordering, unrolling and so on. On the GPU, we utilize the "SSSRRSRS" template to generate the corresponding high-level structure. Here, "S" refers to one tiling level of space loops, and "R" represents one tiling level of reduction loops. The loops in the first three space tilings are bound to BlockIdx, virtual thread (for reducing bank conflicts), and ThreadIdx on the GPU, respectively. Different transformations in the outermost for-loops will result in different features extracted from the innermost assignment statement, leading to varying throughput predictions and learning processes in the cost model. Overall, our method considers various optimizations of the outermost loops during the optimization process. By extracting features from the innermost non-loop statements, we train a learnable cost model on the GPU to automatically optimize the full tensor program. We train the cost model and mainly target the data parallel tensor programs which consist outermost loop-nest and some innermost assignment statements. We use the ATFormer to predict the score of each non-loop innermost statement. An example are shown as follows:
> ```python
> for i in range(10):
>   for j in range(10):
>       B[i][j] = A[i] * 2                          # Statement B
> for i in range(10):
>   	C[i] = B[i][i] - 3                          # Statement C
> ```
> In this example, there are two non-loop innermost statement, B and C. We estimate the cost of these tensor programs. The total cost of it is the sum of cost B and cost C. We perform static analysis to extract the features for every non-loop innermost statement and make prediction for them. These predictions, along with features extracted from the entire program, such as the number of *used cached lines*, *used memory*, *reuse distance*, and *arithmetic intensity curves*  are used to estimate the cost of the entire program. In the following code snippet, we utilize an example of matrix multiplication and ReLU function to elucidate the high-level structure and low-level details as mentioned before.
> ```python
> for i in range(512):
>   for j in range(512):
>     for k in range(1024):
>       C[i, j] += A[i, k] * B[k, j]
> for i in range(512):
>   for j in range(512):
>     D[i, j] = max(C[i, j], 0)
> ```
> One of the possible high-level structure transformation is shown as below:
> ```python
> for i.0 in range(Tile_I0):
>   for j.0 in range(Tile_J0):
>     for i.1 in range(Tile_I1):
>       for j.1 in range(Tile_J1):
>         for k.0 in range(Tile_K0):
>           for i.2 in range(Tile_I2):
>             for j.2 in range(Tile_J2):
>               for k.1 in range(Tile_I1):
>                 for i.3 in range(Tile_I3):
>                   for j.3 in range(Tile_J3):
>                     C[...] += A[...] * B[...]
>         for i.4 in range(Tile_I2 * Tile_I3):
>           for j.4 in range(Tile_J2 * Tile_J3):
>             D[...] += max(C[...], 0.0)
> ```
> The following two code snippets demonstrate the low-level details that can be generated based on the aforementioned high-level structure in the hierarchical search space.
> ```python
> parallel i.0@j.0@i.1@j.1 in range(256):
>   for k.0 in range(32):
>     for i.2 in range(16):
>       unroll k.1 in range(16):
>         unroll i.3 in range(4):
>           vectorize j.3 in range(16):
>             C[...] += A[...] * B[...]
>   for i.4 in range(64):
>     vectorize j.4 in range(16):
>       D[...] = max(C[...], 0.0)
> ```
> ```python
> parallel i.2 in range(16):
>   for j.2 in range(128):
>     for k.1 in range(512):
>       for i.3 in range(32):
>         vectorize j.3 in range(4):
>           C[...] += A[...] * B[...]
> parallel i.4 in range(512):
>   for j.4 in range(512):
>     D[...] = max(C[...], 0.0)
> ```
> By examining the provided code snippets, it becomes evident that throughout the entire process of training and prediction in the cost model, there are considerations for common optimizations such as tiling that affect the outermost for-loops. To provide a more comprehensive understanding of our design and optimization methodology, we will showcase detailed code examples in the appendix. These examples will illustrate the process and incorporate common optimizations to enhance readers' comprehension.
>
> **Q2:** What are the high-level structure and low-level structure in hierarchical feature generation?
>
> **A2:**  Each tuned operator in the neural network can be represented as a loop-nested structure in their implementation and then mapped to the specific hardware accelerators for the efficient inference or training. As for the high-level structure, it represents the coarse-grained statement features and it is labeled as a tag such as dense, softmax or conv2d to describe the type of this operator as the graph-level representation. As for the low-level structure, it represents more details about the implemenation and optimization of the operator such as multi-level tiling, reordering and unrolling as the tensor-level representation. It is worth noting that the concepts of high-level and low-level in the hierarchical search space defined in Section 3.1 of the problem formulation differ from the concepts outlined in Section 3.2  hierarchical feature generation. The former emphasizes the overall transformations of tensor program, while the latter pertains to the representation and extraction of features within the innermost assignment statements. We appreciate the reviewer's valuable feedback and inquiries regarding this matter. In order to facilitate a deeper understanding of our work, we will further emphasize these two aspects in the revised version, ensuring that readers can grasp them more effectively.
>
> **Q3:** In Figure-5, how to incorporate statement features into self-attention and how to update self-attention after scanning the full tensor program?
>
> **A3:** Figure 5 portrays a high-level diagram that showcases the utilization of self-attention for relationship modeling, where the extracted features from the innermost assignment statement are fed as input.  Figure 3 offers a more lucid depiction of the features extracted from the statement and their subsequent incorporation into self-attention. Initially, a feature vector of dimension 164 will be extracted  for the innermost assignment statement including computation, memory access, and arithmetic strength. Subsequently, this vector will undergo concatenation with a 10-dimensional one-hot operator category feature, resulting in the creation of a new feature vector with a length of 174. Next, the embedding layer will embed each innermost assignment statement extracted from different full tensor programs into a new dimensional space representation. Finally, these embeddings will be multiplied by learnable weight matrices to obtain the corresponding Q, K, and V matrices. This comprehensive process enables end-to-end training and facilitates the updating of the self-attention mechanism.
>
> **Q4:** Why is the throughput metric used in the loss function although the experiment results show a latency improvement?
>
> **A4:** Thanks for the comment. The cost model is used to predict the fitness of each tensor program, which is the throughput of one tensor program. During the process of model inference on a single machine with a single GPU, it is generally assumed that there exists an inverse relationship: ${throughput} ∝ \frac{1}{latency}$. It means that as the throughput increases, the inference latency decreases. Therefore, there is a prevailing benchmark standard for performance evaluation of the entire end-to-end network lies in the inference latency in both Academia and industry when developing a new inference engine or framework. Certainly, if throughput is employed exclusively during experimental evaluations, it will yield identical conclusions to those obtained using inference latency as the metric.
>
> **Q5:** Table-4 should include one or two nlp models.
>
> **A5:** Thanks for the comments. Due to the limited rebuttal time, we add the nlp-based model Bert-Tiny in the table below to further verify our design.
> |      Bert-Tiny       |          |          |          |          |          |          |
> | :------------------: | -------- | -------- | -------- | -------- | -------- | -------- |
> |       Methods        | (a)      | (b)      | (c)      | (d)      | (e)      | (f)      |
> |        Mask?         |          |          | &#10004; | &#10004; |          |          |
> |     Pre-trained?     |          |          |          | &#10004; | &#10004; |          |
> |      RMSE Loss?      | &#10004; |          |          |          |          |          |
> |      Rank Loss?      |          | &#10004; | &#10004; | &#10004; | &#10004; | &#10004; |
> |       AutoTVM?       |          |          |          |          |          | &#10004; |
> | Total Latency （ms） | 4.18     | 3.41     | 3.97     | **2.32** | 2.46     | 5.07     |
> |   Search Time (s)    | 1127     | 1141     | 1150     | **818**  | 816      | 3826     |
>
> In the revised version, we will integrate a wider nlp-based models to facilitate the comprehension of our design for the readers.

---

### Official Review · Reviewer_astq · 2023-08-10

**Soundness:** 4

**Excitement:**

4: Strong: This paper deepens the understanding of some phenomenon or lowers the barriers to an existing research direction.

**Paper Topic And Main Contributions:**

In order to optimize deep neural network efficiency and overcome challenges such as long optimization times and deployment transfer to other hardware, this paper proposes a cost modeling and tensor program optimization framework, ATFormer.

The tuning problem is framed as an optimization problem in a hierarchical search space. ATFormer uses attention modules to capture global/long-range dependencies in features.

The model is evaluated in terms of traditional (online) and transfer learning, with results indicating improved latencies and search times across GPU devices.

Other contributions include extensive performance analysis of auto-tuning with different strategies (tree-based/DNN-based approaches).


**Questions For The Authors:**

A) What is method (e) in line 549?

B) Why might performance for LSTM be better on Bert-Tiny and VGG-16? That could be an interesting insight.

C) The batch size is notably set to 1. Are the insights of this paper applicable to larger batch sizes?

D) Are there experimental details/settings for the online datasets? They appear to be dynamic with execution, but it may be helpful to include details of their size/settings for reproducibility.


**Reasons To Accept:**

The research problem of interest (neural network compilation for efficiency) has critical implications for practitioners. The authors evaluate a wide range of DNNs (MobileNet/ResNet/BERT/GPT to name a few) and make a solid, unique effort in this direction with their approach.

The approach is well-motivated based on challenges with past approaches (large search space, cross-device transferability, long optimization times).

There are adequate baselines (XGBoost, LSTM, TabNet) that the approach outperforms on many models, and the ablations are insightful.

The testing across other GPU platforms (NVIDIA 2080 Ti with Turing / NVIDIA 3090 GPU with Ampere) is interesting and practical.


**Reasons To Reject:**

Some of the gains seem to be incremental (Fig. 6); LSTM outperforms ATF on 2 out of the 6 models in inference latency.

One of the primary claims of the paper is that the approach demonstrates effective transfer across devices. While 2 GPUs were tested, evaluating on more GPUs with variance in architecture would strengthen the claims.


**Reproducibility:**

3: Could reproduce the results with some difficulty. The settings of parameters are underspecified or subjectively determined; the training/evaluation data are not widely available.

**Reviewer Confidence:**

3: Pretty sure, but there's a chance I missed something. Although I have a good feel for this area in general, I did not carefully check the paper's details, e.g., the math, experimental design, or novelty.

**Typos Grammar Style And Presentation Improvements:**

There are a fair amount of typos, which warrants further proofreading of the entire paper. Noting some:
- Line 53 “an” should be “can”
- Line 231 “perdiction”
- Line 528 “AFTormer”
- Figure 7 X-axis overlaps

In general, the figures/table text should be larger.

It would help to define ATFormer-1L and ATFormer-M/Mask earlier in paper.

Table 4 and 6 would benefit from a clearer outline of (a) - (f) in the text somewhere.

---

> ### Author Rebuttal · Authors · 2023-08-29
>
> We sincerely thank you for your comments as well as the appreciation of our work. Our responses to your concerns are demonstrated below:
>
> **Q1:** What is method(e) in line 549?
>
> **A1:** Method (e) corresponds to the utilization of ATFormer with two attention blocks (ATFormer-2L) that are trained through transfer learning and rank loss to fine-tune the tensor program. The distinction between method (e) and method (d) lies in the fact that method (d) incorporates a masking mechanism known as ATFormer-M during the training phase.
>
> **Q2:** Why might performance for LSTM be better on Bert-Tiny and VGG-16?
>
> **A2:** Our LSTM model is relatively simple, consisting of only one layer with a hidden dimension of 1024. As a result, it has fewer parameters compared to the ATFormer-series models. In contrast to the variable kernel sizes used in models like ResNet or MobileNet, the VGG-16 network has fixed kernel sizes in its convolutional layers, making its architecture straightforward. This stability in the predictions of LSTM differentiates it from the interweaving of different convolutional layer sizes. Similarly, in the Bert-Tiny model, the dimensions of the optimization matrix for Q, K, and V in the encoder remain unchanged. This characteristic allows LSTM to achieve slightly better performance in terms of cost model updates using online datasets compared to the ATFormer series. Moreover, another potential factor is that the construction of the online dataset is relatively small in scale. When training a cost model from scratch, the presence of fewer parameters in the cost model can facilitate quicker convergence of its performance. However, it should be noted that these improvements are mainly observed in traditional learning with online datasets. In transfer learning experiments, particularly in optimizing NLP-based models, LSTM cost model does not outperform the superior performance achieved by ATFormer cost models, regardless of the total tuning time or final optimized inference latency.
>
> **Q3:** The batch size is notably set to 1. Are the insights of this paper applicable to larger batch sizes?
>
> **A3:** Thanks for pointing out this interesting part. We would like to highlight that our approach is also suitable for scenarios involving large batch sizes. In order to validate our design within the given time constraints during the rebuttal phase, we perform experiments using 8 batches on the NVIDIA 3090 GPU via traditional learning. The experiments are conducted on Bert-Tiny and ResNet-18, utilizing the same baselines and settings as outlined in our original paper.. More details can be found in the following tables.
> | batch = 8   | ResNet-18 (ms) | Bert-Tiny (ms) |
> | ----------- | -------------- | -------------- |
> | XGBoost     | 3.04           | 17.42          |
> | LSTM        | 3.35           | 14.83          |
> | TableNet    | 2.88           | 16.98          |
> | MHA         | 2.79           | 15.32          |
> | ATFormer-1L | 2.51           | 15.49          |
> | ATFormer-2L | **2.39**       | **14.37**      |
>
> In the revised version, we intend to expand our analysis by including additional batch sizes such as 2, 4, and 16 for both transfer learning and traditional learning scenarios. This will provide a more comprehensive understanding of our design across a range of batch sizes.
>
> **Q4:** Are there experimental details/settings for the online datasets? They appear to be dynamic with execution, but it may be helpful to include details of their size/settings for reproducibility.
>
> **A4:** Thanks for the comments. ATFormer is built on top of Ansor[1], which employs an online data collection approach during the search process to construct the datasets. In order to maintain a fair comparison with the baselines, we have continued to employ the same approach as Ansor for constructing online datasets. Our approach also combines an evolutionary search algorithm with iterative construction of online datasets. In each iteration, we utilize newly resampled programs and good programs from previous iterations as the initial population for the evolutionary search. We use a set of evolution operations, including *Tile size mutation*, *Parallel mutation*, *Pragma mutation*, *computation location mutation*, and *node-based crossover*, to rewrite and generate tensor programs. The evolutionary search leverages mutation and crossover to repeatedly generate new candidate programs over multiple rounds, ultimately producing a small set of tensor programs with the highest scores. These selected tensor programs are then compiled and executed on a GPU platform to measure the real inference latency. The collected measurement data is subsequently utilized to update the cost model, gradually enhancing its accuracy to better match the GPU performance with the online datasets. For more detailed information, please refer to the hyperparameters specifically designed for the evolutionary search in collecting the online datasets:
> | Parameters                        | Value    |
> | ---------------------- | -------- |
> | sample_init_min_population        | 50       |
> | sample_init_use_measured_ratio    | 0.2      |
> | evolutionary_search_population    | 2048     |
> | evolutionary_search_num_iters     | 4        |
> | evolutionary_search_mutation_prob | 0.85     |
> | gpu_multi_level_tiling_structure  | SSSRRSRS |
> | max_innermost_split_factor        | 64       |
>
> The term "gpu_multi_level_tiling_structure" denotes the utilization of the "SSSRRSRS" template to generate the high-level structure of tensor programs. In this template, "S" represents one tiling level of space loops, while "R" represents one tiling level of reduction loops. The loops in the first three space tilings are bounded to BlockIdx, virtual thread (to reduce bank conflicts), and ThreadIdx on the GPU, respectively.
>
> To facilitate reproducibility and further development on ATFormer, we will open source all of code including training and testing, as well as pretrained models.
>
> [1] Zheng, Lianmin and Jia, Chengfan and Sun, Minmin and Wu, Zhao and Yu, Cody Hao and Haj-Ali, Ameer and Wang, Yida and Yang, Jun and Zhuo, Danyang and Sen, Koushik and others, 14th USENIX symposium on operating systems design and implementation (OSDI 20)
>
> **Q5:** Typos and figures/tables size comments.
>
> **A5:** Thanks for the comments. We will fix all the typos in the revised version. In the original submission, certain figures and tables were compressed to comply with the page limitation. In the revised version, we will carefully condense some of the content and move it to the appendix, thus creating more appropriate space for these tables and figures.
>
> **Q6:** It would be helpful to define ATFormer-1L/M earlier in paper and clearly outline of (a) - (f).
>
> **A6:**  We totally agree with you and we provide more elaborate descriptions below. The model we refer to as ATFormer-1L is composed of a single layer of attention block, distinguishing it from the standard ATFormer cost model, which encompasses two layers of attention blocks. Furthermore, ATFormer-M indicates the utilization of a mask mechanism during training to fortify the stability and performance of ATFormer. Method (a) refers to training ATFormer using rmse loss via traditional learning to optimize the network's inference performance. Method (b) involves substituting the rmse loss in method (a) with rank loss. Method (c) builds upon method (b) by incorporating ATFormer-M, which integrates the mask mechanism, to further enhance the performance of cost model. Method (d) extends method (c) by integrating transfer learning with pretrained-models to expedite the tuning process of the network. Method (e) combines the standard ATFormer via transfer learning without the mask mechanism for optimization. Method (f) represents the baseline approach using AutoTVM for operator tuning. In this method, only rank loss is employed, and the operator is optimized through traditional learning. Finally, in the revised version, we will integrate these variants into the appropriate sections to facilitate readers' comprehension of our design and configuration.
>
> **Q7:**  While 2 GPUs were tested, evaluating on more GPUs with variance in architecture would strengthen the claims.
>
> **A7:** Thanks for the comments. In the future work, we will extend and develop the ATFormer to a broader range of architectures, including NVIDIA Jetson AGX, Xavier, NX, TX2, as well as the emerging hopper architecture, to accelerate a wider variety of operators and models.

---

### Meta-Review · Area_Chair_ukxf · 2023-09-18

**Recommendation:** 5

**Metareview:**

The paper introduces ATFormer, a cost modeling and tensor program optimization framework for enhancing deep neural network efficiency. By framing the tuning problem as an optimization task in a hierarchical search space, ATFormer employs attention mechanisms to capture global and long-range dependencies in features. This model is assessed both traditionally and through transfer learning.

Most reviewers asked for additional explanation/results related to code generation of the approach. Initially, reviewers interpreted the work to optimize only innermost loops, limiting its applicability. However, the authors provided extensive feedback and better explanation for the method (and example output) in the rebuttal and convinced the reviewers that the approach is a more general framework. I suggest that the authors try to refactor the paper to make this more clear and provide some more interesting examples upfront (taken from the rebuttal discussion) to highlight to what degree this work can perform optimization for tensor programs.

---

### Decision · Program_Chairs · 2023-10-07

**Decision:**

Accept-Main

**Comment:**

The paper introduces ATFormer, a cost modeling and tensor program optimization framework for enhancing deep neural network efficiency. By framing the tuning problem as an optimization task in a hierarchical search space, ATFormer employs attention mechanisms to capture global and long-range dependencies in features. This model is assessed both traditionally and through transfer learning.

Most reviewers asked for additional explanation/results related to code generation of the approach. Initially, reviewers interpreted the work to optimize only innermost loops, limiting its applicability. However, the authors provided extensive feedback and better explanation for the method (and example output) in the rebuttal and convinced the reviewers that the approach is a more general framework. I suggest that the authors try to refactor the paper to make this more clear and provide some more interesting examples upfront (taken from the rebuttal discussion) to highlight to what degree this work can perform optimization for tensor programs.